# Self-evolving photonic crystals for ultrafast photonics

Takuya Inoue [1,3] ✉, Ryohei Morita[2,3], Kazuki Nigo[1], Masahiro Yoshida [2], Menaka De Zoysa[1], Kenji Ishizaki[1] & Susumu Noda [1] ✉

Ultrafast dynamics in nanophotonic materials is attracting increasing attention from the perspective of exploring new physics in fundamental science and expanding functionalities in various photonic devices. In general, such dynamics is induced by external stimuli such as optical pumping or voltage application, which becomes more difficult as the optical power to be controlled becomes larger owing to the increase in the energy required for the external control. Here, we demonstrate a concept of the self-evolving photonic crystal, where the spatial profile of the photonic band is dynamically changed through carrier-photon interactions only by injecting continuous uniform current. Based on this concept, we experimentally demonstrate short-pulse generation with a high peak power of 80 W and a pulse width of <30 ps in a 1-mm-diameter GaAs-based photonic crystal. Our findings on self-evolving carrier-photon dynamics will greatly expand the potential of nanophotonic materials and will open up various scientific and industrial applications.

Ultrafast control of optical phenomena inside nanophotonic materials such as photonic crystals and metamaterials[1-10] is attractive for both fundamental physics and industrial applications. In order to realize such dynamic control, it is necessary to induce a change of the refractive index or absorption coefficient which is strong enough to alter their optical dispersions. Although such dynamic control was realized by optical pulse irradiation[1,3,4,7] or electrical pulse application[5,6,8] in previous studies, the former requires bulky external high-power lasers and the latter is difficult to realize on ultrafast (<ns) time scales. These problems become more and more serious as the device size increases and the optical power to be controlled becomes larger owing to the increase in the required energy for the external control. Other methods such as the use of temperature-sensitive phase-change materials[9,10] or micro-electro mechanical systems[2,6] have been also investigated, but they suffer from much slower response speeds.

Here, to overcome these issues involved in external control, we propose a concept of the self-evolving photonic crystal, where the spatial profile of the photonic band structure is dynamically changed without any external ultrafast stimulus. This concept is based on an ultrafast change of the refractive index induced by stimulated emission inside photonic-crystal surface-emitting lasers (PCSELs)[11-15], leading to spontaneous short-pulse generation with a high peak power of 80 W and a pulse width of <30 ps.

## Results

### Principle of self-evolving photonic crystal

The schematic of the proposed self-evolving photonic crystal is shown in Fig. 1a. Here, the lattice constant is gradually increased along the $u$-axis as shown in the right panel. The cross section of the structure is shown in the left panel, where the above graded photonic crystal is located near the active layer and is incorporated inside a p-n junction for current injection. For the photonic-crystal layer, we employ a double-lattice photonic crystal, in which two holes are shifted in the $x$ and $y$ directions by about one quarter of the lattice constant[15,16] (see Supplementary Note 1 for details).

Figure 1b shows the photonic band structure of a typical double-lattice photonic crystal with a given lattice constant. As detailed in our previous paper[16], for a double-lattice photonic crystal which has reflection symmetry along the $u$-axis, the

[1]Photonics and Electronics Science and Engineering Center, Kyoto University, Kyoto, Japan. [2]Department of Electronic Science and Engineering, Kyoto University, Kyoto 615-8510, Japan. [3]These authors contributed equally: Takuya Inoue, Ryohei Morita. ✉e-mail: t_inoue@qoe.kuee.kyoto-u.ac.jp; snoda@kuee.kyoto-u.ac.jp

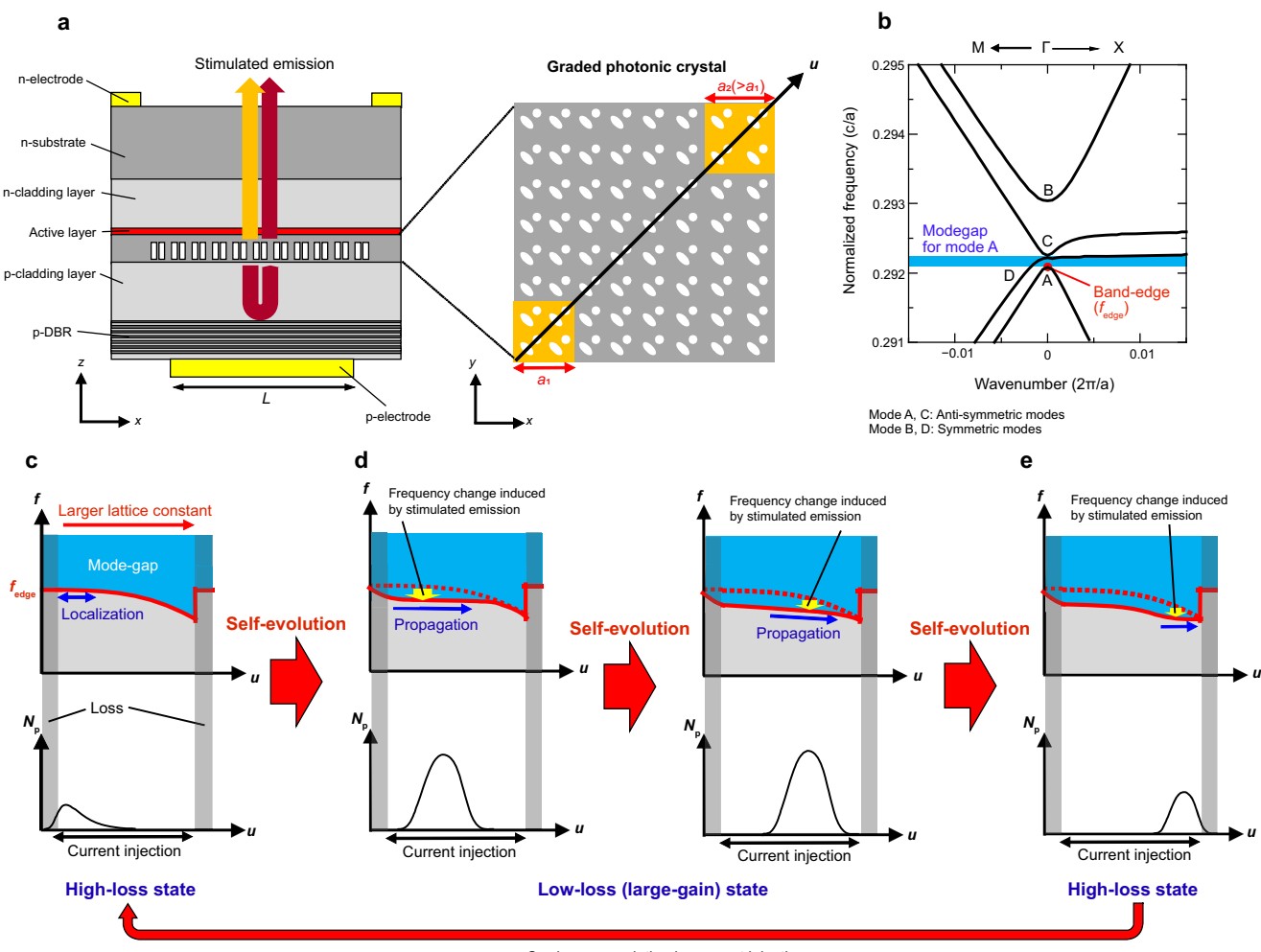

**Fig. 1 | Principle of self-evolving graded photonic crystals for short-pulse generation. a** Schematic picture of a photonic-crystal surface-emitting laser (PCSEL) with a self-evolving graded photonic-crystal, where the lattice constant of a double-lattice photonic crystal is monotonically increased inside the current injection region. **b** Typical photonic band diagram of a double-lattice photonic crystal. **c** Band-edge frequency distribution $f_{edge}$ (upper) and photon density distribution $N_p$ (lower) inside the self-evolving photonic crystal before lasing. **d** Band-edge frequency distributions (upper) and photon density distributions (lower) inside the self-evolving photonic crystal during lasing. Self-evolution of band-edge frequency distribution is caused by stimulated-emission-induced refractive-index change, which enables short-pulse high-power pulse generation. **e** Band-edge frequency distribution (upper) and photon density distribution (lower) inside the self-evolving photonic crystal at the end of lasing.

band-edge modes can be classified into the following two groups: (1) anti-symmetric modes (A, C), which have electric-field vectors that are anti-symmetric about the $u$-axis, and (2) symmetric modes (B, D), which have electric-field vectors that are symmetric about the $u$-axis. In conventional PCSELs with uniform lattice constants, a band-edge mode with the smallest radiation constant (mode A in this case) induces a two-dimensional standing-wave resonance spreading over the whole area of the current injection region, resulting in uniform coherent lasing. On the other hand, in the proposed graded photonic crystal, the lattice-constant gradation induces the gradation of the photonic band-edge frequency as shown in Fig. 1c, which initially prevents the resonant mode from spreading over a large area due to the existence of the photonic mode-gap shown with blue in Fig. 1b (it should be noted that band D does not affect mode A due to the difference of the electric-field symmetry as described above). The resonant mode is repelled toward the outside of the current injection region as shown in the lower panel of Fig. 1c, where the modal loss increases because the active layer outside the current injection region induces absorption rather than gain. Once the device starts to lase, the band-edge frequency gradation where the photons exist is dynamically

compensated by refractive-index change induced by stimulated emission (or stimulated carrier recombination), as shown in the upper panels of Fig. 1d. As a result, reflection due to the mode gap is weakened and the light propagates into the neighboring section in which many carriers are accumulated, leading to further amplification of light as shown in the lower panels of Fig. 1d. Such ultrafast self-evolution of the photonic band causes the spontaneous transition from high-loss to low-loss (large-gain) states and enables high-power short-pulse generation just by injecting continuous uniform current into the single electrode. When the photons reach the other side of the edge (Fig. 1e), lasing oscillation halts because the carriers accumulated inside the entire current injection section are consumed, returning to a high-loss state. The above process of self-evolution is repeated many times as long as the current is supplied into the device.

## Numerical simulations

To confirm the above-mentioned principle of self-evolution, we performed a numerical simulation of the transient waveforms of the proposed device by time-dependent three-dimensional coupled-wave theory[17]. The details of this simulation are provided in Supplementary

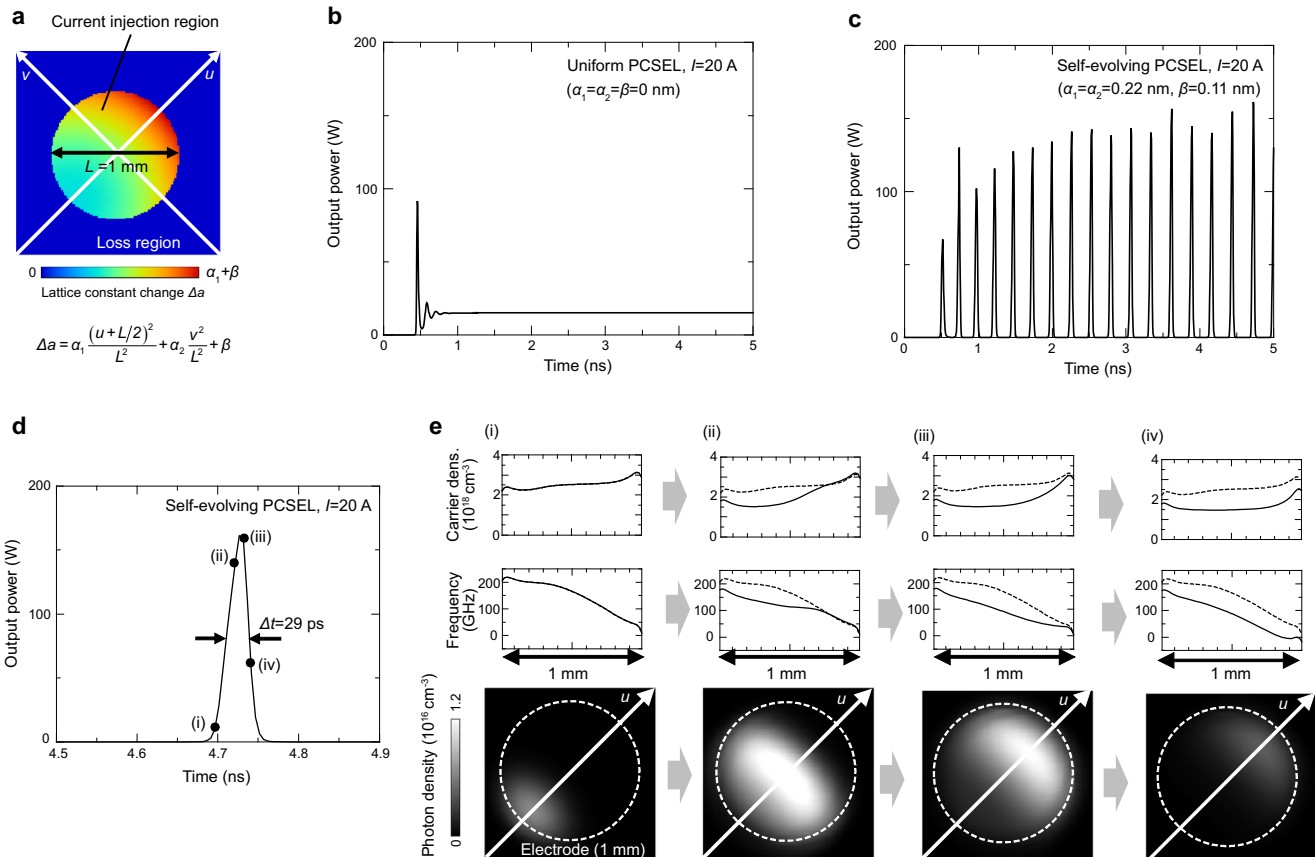

**Fig. 2 | Simulated transient response of self-evolving photonic crystals.**
**a** Simulation model of a self-evolving photonic crystal with 1-mm-diameter current injection. **b, c** Transient response without and with band-edge frequency gradation. A constant injection current of 20 A is used in the simulation. **d** Enlarged view of a single pulse in **c**. **e** Carrier-density distributions, band-edge frequency distributions, and photon density distributions at four different timings (i)–(iv) in **d**. Dashed lines in panels (ii)–(iv) show the distributions at the initial stage of lasing (i).

Note 2. In this simulation, we used a double-lattice photonic crystal composed of pairs of elliptical and circular air holes as shown in Fig. 1a. The distance between the holes of each pair and their filling factors were appropriately adjusted to achieve a moderate radiation constant of the lasing mode ($\alpha_v \sim 10$ cm$^{-1}$) and a large threshold margin between the fundamental mode and higher-order modes ($\Delta\alpha_v \sim 9$ cm$^{-1}$)[16] (see Supplementary Note 1). To introduce the band-edge frequency gradation, the lattice constant ($a$) inside the current injection region was gradually changed using three gradient parameters ($\alpha_1$, $\alpha_2$, $\beta$) while that in the surrounding region was fixed, as shown in Fig. 2a. Here, $\alpha_1$ represents the maximum lattice constant difference along the $u$-axis of the current injection region, which physically determines the magnitude of the mode-gap effect shown in Fig. 1c, while $\beta$ represents the lattice constant difference inside and outside the current injection region, which determines the in-plane loss of the lasing mode outside of the current injection region (see Supplementary Note 3 for details). In the designed graded photonic crystal, a lattice constant gradation along the $v$-axis ($\alpha_2$) is also considered. Such biaxial gradation can compensate the carrier-induced refractive-index distribution due to spatial hole burning along the $v$-axis, in order to greatly narrow the beam divergence angle (see Supplementary Note 4 for details). Figure 2b, c show the calculated temporal waveforms of the output power at an injection current of 20 A for devices without and with frequency gradation under uniform current injection. When the photonic crystal is uniform ($\alpha_1 = \alpha_2 = \beta = 0$ nm, Fig. 2b), a constant output power is obtained after relaxation oscillations, which corresponds to single-mode continuous-wave lasing. On the other hand, in the self-evolving graded photonic crystal ($\alpha_1 = \alpha_2 = 0.22$ nm, $\beta = 0.11$ nm, Fig. 2c), the

temporal waveform changes to intermittent short-pulse trains whose peak powers are >100 W. Figure 2d shows an enlarged view of a single pulse in Fig. 2c, where a pulse width of ~30 ps obtained just by uniform constant current injection is confirmed. To visualize the mechanism of the short-pulse generation, Fig. 2e shows the calculated carrier-density distributions, band-edge frequency distributions, and photon density distributions inside the device at four different timings during the pulse generation in Fig. 2d. As shown in these figures, the resonant mode localizes at the edge of the current injection section at the initial stage of lasing owing to the pre-designed band-edge frequency gradation and it moves to the center of the device as the slope of the band-edge frequency gradation decreases owing to the carrier-induced refractive index change during the pulse amplification, which verifies the principle of self-evolution shown in Fig. 1. As detailed in Supplementary Note 3, the above operation is robustly obtained over a wide range of injection currents and gradient parameters ($\alpha_1$, $\alpha_2$, $\beta$) and even when the random fluctuations of the band-edge frequencies exist. It should be also noted that the thermally induced refractive-index change of the device, if any, can be considered as static because the time constant of the temperature change of the device (several microseconds) is 4–5 orders of magnitude slower than that of the self-evolving effect (several tens of picoseconds). Therefore, such thermal effect can be compensated by the adjustment of the pre-designed gradient parameters. In addition, by controlling the temperature distribution of the device via the control of the current injection distribution, it might be also possible to effectively realize a graded photonic crystal to achieve self-evolution even without the pre-designed lattice constant distribution.

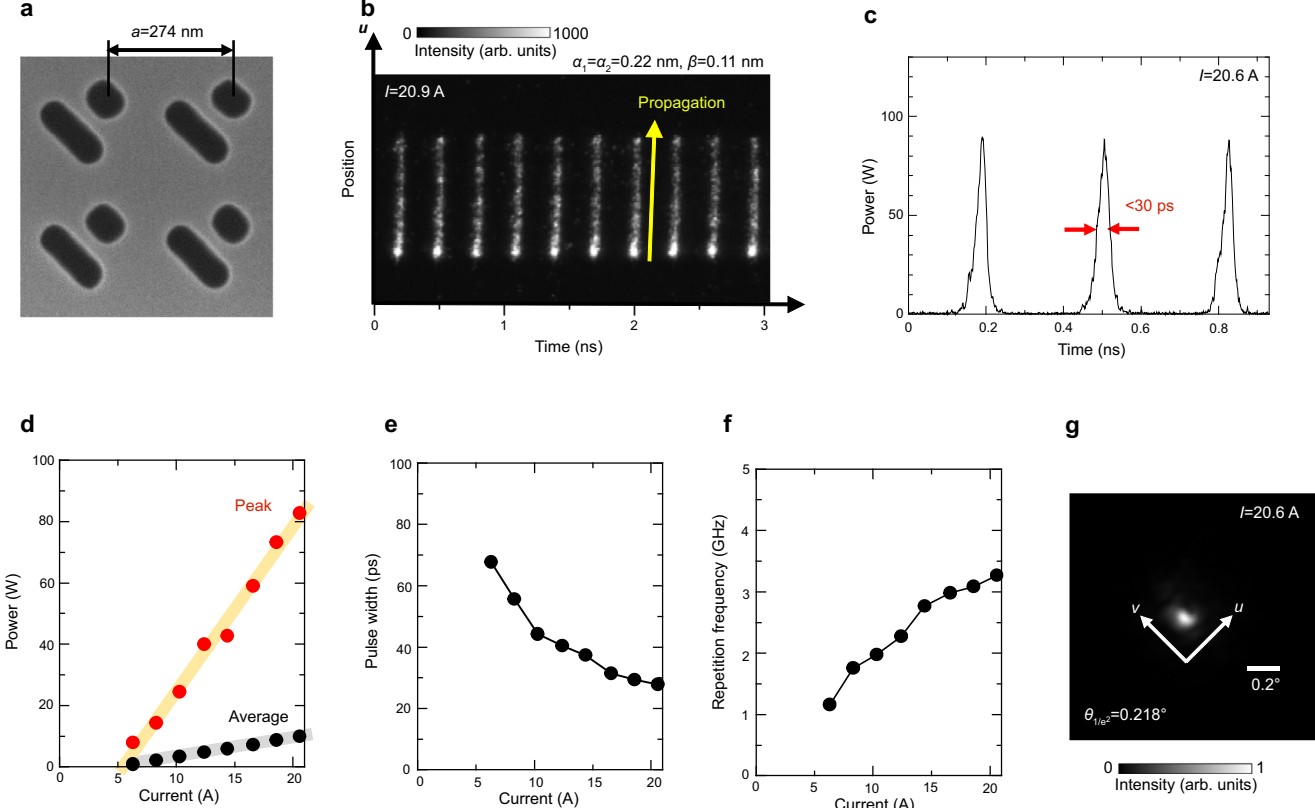

**Fig. 3 | Experimental demonstration of short-pulse generation in self-evolving photonic crystals. a** Scanning electron microscope image of a fabricated double-lattice photonic crystal. **b** Streak camera image of the fabricated device at an injection current of 20.9 A. **c** Temporal change of the output power of the device at an injection current of 20.6 A. **d** Peak power and average power of the device as a function of injection current. Orange and gray lines are drawn for visual guidance. **e** Pulse width as a function of injection current. **f** Pulse repetition frequency as a function of injection current. **g** Measured far-field beam pattern at an injection current of 20.6 A. $\theta_{1/e^2}$ is the average value of the divergence angles evaluated at $1/e^2$ of the maximum in the $x$ and $y$ directions.

## Experimental demonstrations

We fabricated the designed 1-mm-diameter self-evolving photonic crystal to demonstrate high-peak-power short-pulse lasing based on self-evolution. The fabrication process is the same as that of conventional PCSELs, the details of which are explained in the Methods section. It should be emphasized that the proposed device requires no multi-section electrodes nor saturable absorbers, which are required for conventional $Q$-switched semiconductor lasers[18–23]. A scanning electron microscope (SEM) image of the fabricated double-lattice photonic crystal is shown in Fig. 3a. The average lattice constant of the fabricated photonic crystal is 274 nm, which corresponds to a lasing wavelength of 936 nm. In the experiment, we fabricated two devices, one with monoaxial gradation ($\alpha_1 = 0.22$ nm, $\alpha_2 = 0$ nm, $\beta = 0.11$ nm) and the other with biaxial gradation ($\alpha_1 = \alpha_2 = 0.22$ nm, $\beta = 0.11$ nm). Since the magnitude of the designed lattice constant change ($\Delta a$) is <0.5 nm, it is impossible to directly observe the lattice-constant change even using the SEM. However, as shown in the following equation, the positional shift of each lattice point from its original position [$\Delta x(m, n)$, $\Delta y(m, n)$, where $m$ and $n$ are positive integers that denote the $x$ and $y$ coordinates of the point] becomes several tens to hundreds of nanometers after the summation of $\Delta a$ over many periods, which enables the fabrication of the designed sub-nanometer lattice-constant gradation.

$$\Delta x(m, n) = \sum_{m'=1}^{m} \Delta a(m', n), \quad \Delta y(m, n) = \sum_{n'=1}^{n} \Delta a(m, n') \quad (1)$$

For the characterization of the fabricated device, we coupled the laser beam emitted from the device into a slit of a streak camera, and we measured the spatial-temporal evolution of the laser beam. The slit of the streak camera is parallel to the $u$-axis of the graded photonic crystal shown in Fig. 2a. The other details of the characterization are provided in the Methods section. The measured streak camera image of the fabricated graded device with biaxial gradation ($\alpha_1 = \alpha_2 = 0.22$ nm, $\beta = 0.11$ nm) at an injection current of 20.9 A is shown in Fig. 3b, where periodic pulse trains are observed. The camera image for the device with monoaxial gradation ($\alpha_2 = 0$ nm) is shown in Supplementary Note 4. In each pulse, lasing starts from one edge and then propagates in the $+u$ direction, in agreement with the simulated results shown in Fig. 2e. The measured streak camera images for other injection currents are provided in Supplementary Note 5, where stable pulsation is obtained over a wide range of injection currents (8 A – 20 A). The temporal change of the output power at 20.6 A after spatial integration is shown in Fig. 3c, where self-pulsation with a pulse width of as short as <30 ps is successfully obtained. Figure 3d–f show the injection current dependence of the output power (peak and average), pulse width, and repetition frequency, respectively. As shown in Fig. 3d, we obtained a maximum peak power of >80 W, which is four times larger than the highest peak power ever obtained among all $Q$-switched semiconductor lasers without amplifiers[22,23]. The experimental peak power was smaller than the simulated peak power plotted in Fig. 2, which was likely due to differences between the radiation constant and injection current uniformity of the fabricated device and those assumed in the calculations, resulting in different slope efficiencies between the fabricated and simulated devices.

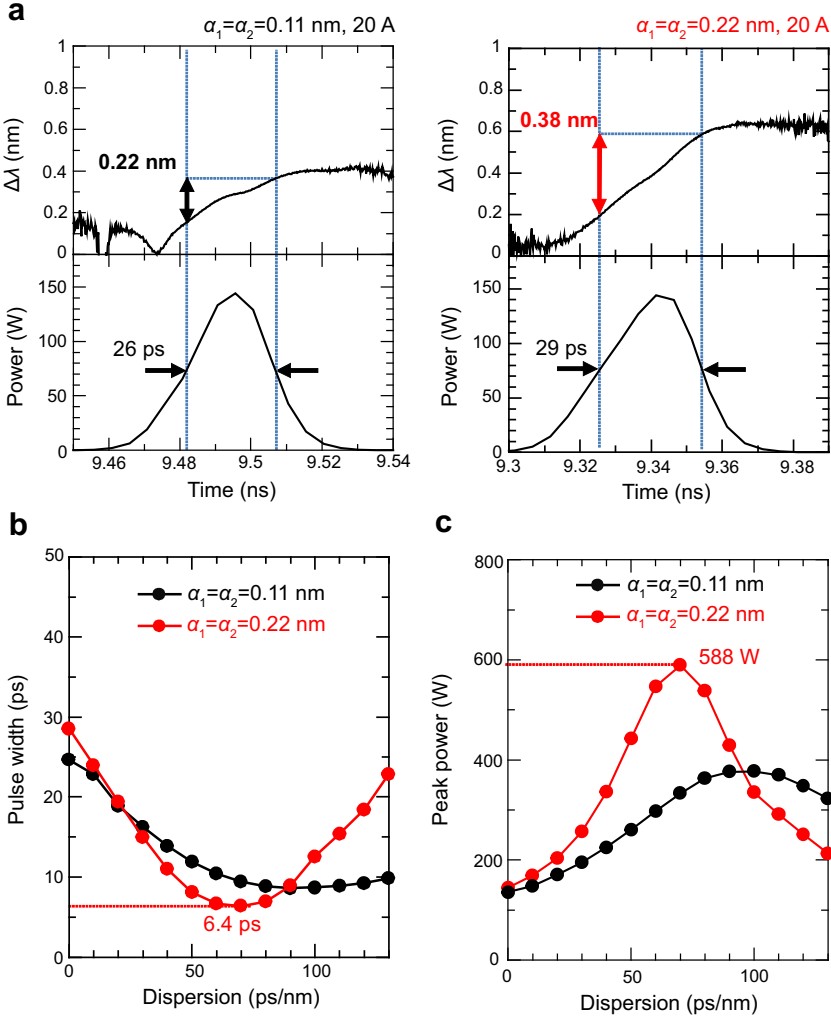

**Fig. 4 | Spontaneous wavelength chirping and pulse compression of self-evolving photonic crystals. a** Calculated instantaneous wavelength change during each pulsation in two self-evolving photonic crystals with different gradient parameters ($\alpha_1 = \alpha_2 = 0.11$ nm and $\alpha_1 = \alpha_2 = 0.22$ nm). The value of $\beta$ is fixed to 0.11 nm in both devices, and an injection current of 20 A is used throughout. **b, c** Calculated pulse width and peak power of the self-pulsation for the devices shown in **a** as a function of the magnitude of second-order dispersion compensation.

As shown in Fig. 3e, f, the pulse width decreases and the repetition frequency increases as the injection current increases. These results are ascribed to the faster carrier accumulation and the faster band-edge frequency change inside the self-evolving photonic crystal at higher injection currents; these experimental results agree well with the simulated results shown in Supplementary Figure S2. The possibility of the external control of the repetition frequency via the superimposition of a radio frequency signal is discussed in Supplementary Note 6. The measured far-field pattern at 20.6 A is shown in Fig. 3g, where a narrow divergence angle of $\theta_{1/e^2} \sim 0.2°$ was observed. The measured beam pattern is slightly elongated along the $v$-axis, which indicates incomplete compensation of the carrier-induced refractive-index distribution along the $v$-axis in the fabricated device (details are explained in Supplementary Note 4). These results confirm that high-peak-power short-pulse lasing based on the self-evolving photonic crystal was successfully realized.

## Discussion

Finally, we discuss the possibility of the generation of shorter-width pulses with higher-peak power via pulse compression in our self-evolving photonic crystal. As shown in Fig. 1, the lasing frequency (wavelength) of our device dynamically decreases (increases) during each pulsation owing to the self-evolution of the band-edge frequency

gradation. By harnessing such a large wavelength chirp with simple dispersion compensation, we can achieve the generation of shorter-width and higher-power pulses. Figure 4a shows the calculated instantaneous wavelength change during each pulsation in two devices with different gradient parameters ($\alpha_1 = \alpha_2 = 0.11$ nm and 0.22 nm). Here, the wavelength of the lasing mode almost linearly increases during pulsation in both devices, and a larger wavelength chirp is obtained for the device with a larger frequency gradation. Figure 4b, c shows the calculated pulse width and peak power of the self-pulsation after employing second-order dispersion compensation as a function of the magnitude of dispersion (the simulation method is detailed in Supplementary Note 7). By optimizing the magnitude of dispersion compensation, we can realize a pulse width of 6.4 ps with a peak power of 588 W in the device with a larger gradation ($\alpha_1 = \alpha_2 = 0.22$ nm); this pulse width is only slightly larger than the Fourier-limited pulse width ($\Delta t = 3.4$ ps) calculated by assuming a Gaussian pulse with a wavelength chirp of $\Delta\lambda = 0.38$ nm. The 6.4-ps pulse width might be further reducible by optimizing the band-edge frequency gradation to suppress the effects of third- and higher-order dispersions. In addition, by increasing the diameter of the photonic crystal and the gradient parameters ($\alpha_1, \alpha_2$), we can realize an even higher-peak power (>1 kW) following dispersion compensation (see details in Supplementary Note 8).

In conclusion, we have proposed and demonstrated the concept of the self-evolving photonic crystal, where the spontaneous transition from a high-loss state to a low-loss state is induced by dynamic evolution of the band-edge frequency distributions inside the device. Our proposed concept brings deeper insights into carrier-photon dynamics inside wavelength-scale photonic nanostructures, and it will inspire various areas of fundamental research on nanophotonic materials, condensed-matter physics, laser materials, and non-linear optics. Based on this concept, we have experimentally realized 80-W-class, 30-ps-width self-pulsation in a 1-mm-diameter self-evolving photonic crystal, and we have also numerically predicted even higher-peak-power, shorter-width pulse generation via simple dispersion compensation. Such one-chip high-peak-power semiconductor lasers will benefit a number of state-of-the-art laser applications such as laser remote sensing[24,25], material processing[26,27], and non-linear laser imaging[28,29].

## Methods

### Device fabrication

First, we prepared an n-GaAs substrate and grew an n-AlGaAs cladding layer, an active layer (InGaAs/AlGaAs triple quantum wells), an AlGaAs carrier blocking layer and an undoped GaAs layer. Next, we deposited a SiN$_x$ hard mask for plasma etching onto the GaAs layer, and then we transferred the graded double-lattice photonic crystal pattern onto this mask by EB lithography and SF$_6$ inductively coupled plasma (ICP) etching. Afterward, we transferred the photonic crystal pattern onto the GaAs layer by etching air holes via BCl$_3$/Cl$_2$/Ar ICP etching. Then, we removed the SiN$_x$ mask by using BHF and the surface oxide by soaking the sample in HCl. After these surface treatments, we grew a p-AlGaAs cladding layer, a p-type distributed Bragg reflector (DBR), and a p$^+$-GaAs contact layer by MOVPE regrowth to embed the photonic crystal pattern inside the device. Finally, we deposited a circular p-electrode onto the p$^+$-GaAs contact layer and a ring-window n-electrode onto the n-GaAs substrate.

### Device characterization

We measured the spatial-temporal evolution of our device using a single-shot streak camera with a temporal resolution of 4 ps (C5680-04, Hamamatsu Photonics K.K.). To avoid any thermal effect, we excited the devices with an electric pulse current with a duration of 150 ns, which is sufficiently longer than the expected period (<1 ns) of the optical pulse trains and can be regarded as direct current. By using two lenses, we transferred the near-field emission pattern of the devices onto the slit of the streak camera, which is parallel to the $u$-axis of the graded photonic crystal. The pulse width and repetition frequency shown in Fig. 3 were obtained by averaging the results of >10 measurements. The average power during current injection was measured by an optical power meter, and the peak power was obtained from the measured temporal waveforms and the measured average power.

### Reporting summary

Further information on research design is available in the Nature Research Reporting Summary linked to this article.

## Data availability

The data that support the plots within this paper and other findings of this study are available within this article and its Supplementary Information file, and are also available from the corresponding author upon request.

## Code availability

The mathematical formulae of time-domain 3D-CWT simulations are available within the Supplementary Information file, and their associated codes are available from the corresponding author upon request.

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

## Acknowledgements

This work was partially supported by a Grant-in-Aid for Scientific Research [20H02655 (T.I.), 22H04915 (S.N.)] from the Japan Society for the Promotion of Science (JSPS), and was also carried out under the project of Council for Science, Technology and Innovation (CSTI), Cross ministerial Strategic Innovation Promotion Program (SIP), "Photonics and Quantum Technology for Society 5.0" (S.N.) and under the CREST program (JP MJCR17N3) commissioned by the Japan Science and Technology Agency (S.N.). The authors thank John Gelleta for fruitful discussions.

## Author contributions

S.N. supervised the entire project with T.I. T.I. designed the devices with K.N. R.M. fabricated the samples with K.N., M.Y., M.D.Z. and K.I. R.M. and K.N. performed the experiments and analysed the data with T.I. T.I and S.N discussed the results with K.N., R.M., M.Y., and wrote the paper.

## Competing interests

The authors declare no competing interests.
