## [Peer Review File · Nature Communications]

Self-evolving photonic crystals for ultrafast photonicsREVIEWER COMMENTS

Reviewer #1 (Remarks to the Author):

The authors propose and demonstrate a new method of ultrafast pulse generation, by utilizing the dynamics in a current-injected photonic crystal surface-emitting laser, in which the equivalent optical cavity has been altered by the refractive index changing induced by carrier accumulation. The authors realize a high peak power of 80W with a pulse width shorter than 30ps, which are indeed quite promising results that are useful for many branches of photonics as the authors claimed. In my opinion, the presented new method is elegant and practical, therefore I believe the novelty of this manuscript fully meets the criteria of NC.

Concerning technical content, here are some comments that hope the authors can address in the revision.

(1) Seems the self-evolving behavior of lasing oscillation depends on the accumulation of carriers that modify the refractive index. Given that other factors, such as the thermal effect will also modify the refractive index, does the proposed method have some robustness to temperature variation? Since the laser operates under high-power status under large current injection, I suppose the thermal effect would be significant in such a design. Hope the authors can discuss and clarify.

(2) The circulation of pulse generation is working as a trigger and release manner, namely, the timing of when the pulse is emitted would depend on the condition of carrier accumulation, which seems not rigorous periodic in time. Could the authors comment on if it is possible the emitting of the pulse can be externally triggered, for instance, by electrical or optical signals, to make the emitting fully periodic?

(3) The authors apply a considerably large PhC (~1mm) for lasing and mentioned that they carefully designed the gradient change for their purpose. In my view, some fabrication defects would make the light localized in a relatively small area that push the device operates as a random laser. Could the authors evaluate the fabrication precision required by their method?

Reviewer #2 (Remarks to the Author):

In this work, the authors propose and demonstrate the spontaneous generation of light pulses by a continuously pumped laser. This result is based on the dynamic modification, induced by the evolution of the charge carrier density, of the band structure of a graded photonic crystal forming the laser cavity. A very complete model, developed in a previous paper by the same authors, allows to simulate precisely this phenomenon that they clearly observe on PCVCSELS lasers whose gain region is constituted of III-V quantum wells. Powerful and short pulses are achieved and additional designs are proposed to improve further these characteristics.

The proposed concept is, from my point of view, quite original and offers a very interesting alternative to the traditional Q-switched lasers since it allows to gather in the same laser cavity all the functionalities necessary for the generation of pulses. This is a very solid work with detailed theoretical analysis, simulation results and experimental demonstrations. I recommend publication. However, I would like to suggest the following changes or additions:

- the authors use the expression "carrier consumption" (see for example top of page 4). I find this expression unclear. Can they clarify?
- we can see on fig1b that the laser mode at the A-band edge has the same energy as a D-band mode that crosses the gap. Can this affect the confinement of the laser mode? Can the authors discuss this point?
- In figure 1c a width "a" is indicated. What does it correspond to? Is it the same width as "L" in figure 1a?
- the evolution of the pulse width and their repetition frequency as a function of various parameters (injection current,...) is studied theoretically (notably in the supplemental). However, it does not seem obvious to me, from the model, what are the optimum values that can be obtained? What are the physical parameters that limit these values? Can this point be commented on?
- the model developed in this article allows to calculate the distribution of charge carriers in the cavity (see for example figure S4 of the supplement). I think it would be interesting to plot this distribution (along u) for the different stages of the pulse propagation (figures 1c-e).
- Apart from simplicity, are there any other advantages to this configuration over traditional Q-switched lasers?

Response to the reviewers' comments (ID: NCOMMS-22-36648-T)

We are grateful to the two reviewers for their positive evaluation of our work and their useful suggestions that have helped us to improve our paper. As indicated in the response that follows, we have addressed all the concerns and suggestions of the reviewers in the revised version of our paper (Manuscript_revised_inoue.docx). The newly included sentences are shown in blue in the revised manuscript.

Reply to Reviewer #1

General Comment

The authors propose and demonstrate a new method of ultrafast pulse generation, by utilizing the dynamics in a current-injected photonic crystal surface-emitting laser, in which the equivalent optical cavity has been altered by the refractive index changing induced by carrier accumulation. The authors realize a high peak power of 80W with a pulse width shorter than 30ps, which are indeed quite promising results that are useful for many branches of photonics as the authors claimed. In my opinion, the presented new method is elegant and practical, therefore I believe the novelty of this manuscript fully meets the criteria of NC.

Reply

We are grateful to the reviewer for his/her positive evaluation of our work. We are greatly encouraged by the reviewer's comment.

Comment 1

Seems the self-evolving behavior of lasing oscillation depends on the accumulation of carriers that modify the refractive index. Given that other factors, such as the thermal effect will also modify the refractive index, does the proposed method have some robustness to temperature variation? Since the laser operates under high-power status under large current injection, I suppose the thermal effect would be significant in such a design. Hope the authors can discuss and clarify.

Reply

We thank the reviewer for his/her important comment. As we have explained in the method section, we performed our measurement with an electric pulse current with a short duration of 150 ns, and thus the thermal effect can be neglected in our experiment. If we further increase the duration of the current injection, the temperature change of the device will become larger and the band-edge

frequency gradation might be changed as the reviewer has pointed out. However, such a thermal effect can be considered as “static” because the time constant of the temperature change of the device (several microseconds) is 4-5 orders of magnitude slower than that of the self-evolving effect (several tens of picoseconds). Therefore, the thermally induced band-edge frequency change, if any, can be compensated by the adjustment of the pre-designed gradient parameters, as proposed in our previous paper [R1]. In addition, by controlling the temperature distribution of the device via the control of the current injection distribution, it might be also possible to effectively realize a “graded” photonic crystal to achieve self-evolution even without the pre-designed lattice constant distribution. In the revised version of the main text (the last paragraph of “Numerical simulations” section), we have briefly included the above discussion on the thermal effect.

[R1] S. Katsuno *et al*, “Self-consistent analysis of photonic-crystal surface-emitting lasers under continuous-wave operation,” *Opt. Express* **29**, 25118 (2021).

Comment 2

The circulation of pulse generation is working as a trigger and release manner, namely, the timing of when the pulse is emitted would depend on the condition of carrier accumulation, which seems not rigorous periodic in time. Could the authors comment on if it is possible the emitting of the pulse can be externally triggered, for instance, by electrical or optical signals, to make the emitting fully periodic?

Reply

We thank the reviewer for his/her insightful comment. As the reviewer points out, the repetition frequency of our device is determined by the amount of carrier consumption in one pulsation and the carrier re-accumulation speed, which depends on various parameters such as the injection current and carrier lifetime. To investigate the possibility of the external triggering of our device, we performed the transient analysis of our graded photonic crystal under the driving condition where a sinusoidal radio frequency (RF) signal is superimposed on direct current (DC). Figure R1a shows the calculated temporal waveforms of the graded photonic crystal designed in the main text ($\alpha_1=\alpha_2=0.22$ nm, $\beta=0.11$ nm) under DC of 20A with a RF signal of $10A_{pp}$ with various frequencies f_{RF} . As shown in the figure, the repetition frequency of the self-pulsation can be externally controlled by the superimposition of RF signals. Figure R1b shows the relationship between f_{RF} and the repetition frequency of the self-pulsation, where both frequencies almost coincide with each other (more precise locking of the repetition frequency may be also possible by replacing the sinusoidal RF signal with another pulsed trigger signal). In the revised version of our manuscript, we have included the above discussion in

Supplementary Section 6.

Fig. R1. a, Calculated temporal waveforms of self-evolving PCSELS under DC of 20A with sinusoidal RF signals with various frequencies f_{RF} . **b**, Relationship between f_{RF} and the repetition frequency of the self-pulsation.

Comment 3

The authors apply a considerably large PhC (~1mm) for lasing and mentioned that they carefully designed the gradient change for their purpose. In my view, some fabrication defects would make the light localized in a relatively small area that push the device operates as a random laser. Could the authors evaluate the fabrication precision required by their method?

Reply

We thank the reviewer for his/her important comment. To quantitatively investigate the robustness of our device against the fabrication defects, we performed the transient analysis of our self-evolving photonic crystal with random fluctuations of the lattice constant distribution. Figures R2a-R2d show the schematics of the lattice-constant distributions ($\alpha_1=\alpha_2=0.22\text{ nm}$, $\beta=0.11\text{ nm}$) with random Gaussian fluctuations with a correlation length of $20\text{ }\mu\text{m}$ and various standard deviations ($\sigma_{fluc}/\alpha_1=0\%$, 3% , 6% , 9%). Figures R2e-R2h show the calculated temporal waveforms of the designed photonic crystals at 20A. The stable pulse generation is maintained even when a random Gaussian fluctuation of $\sigma_{fluc}/\alpha_1=6\%$ is superimposed on the designed lattice-constant distribution. It should be noted that the error of our fabrication process should be smaller than the above value judging from the experimental realization of stable pulse generation. In the revised version of our manuscript, we have included the above discussion on the robustness of our device in the last paragraph of Supplementary Section 3.

Fig. R2. **a-d**, Schematics of lattice-constant distributions ($\alpha_1=\alpha_2=0.22$ nm, $\beta=0.11$ nm) of self-evolving photonic crystals with random Gaussian fluctuations with a correlation length of $20\ \mu\text{m}$ and various standard deviations ($\sigma_{\text{fluc}}/\alpha_1=0\%$, 3% , 6% , 9%). **e-h**, Calculated temporal waveforms of the designed photonic crystals with fluctuations at 20\AA .

Reply to Reviewer #2

General Comment

In this work, the authors propose and demonstrate the spontaneous generation of light pulses by a continuously pumped laser. This result is based on the dynamic modification, induced by the evolution of the charge carrier density, of the band structure of a graded photonic crystal forming the laser cavity. A very complete model, developed in a previous paper by the same authors, allows to simulate precisely this phenomenon that they clearly observe on PCVCSELS lasers whose gain region is constituted of III-V quantum wells. Powerful and short pulses are achieved and additional designs are proposed to improve further these characteristics.

The proposed concept is, from my point of view, quite original and offers a very interesting alternative to the traditional Q-switched lasers since it allows to gather in the same laser cavity all the functionalities necessary for the generation of pulses. This is a very solid work with detailed theoretical analysis, simulation results and experimental demonstrations. I recommend publication.

Reply

We are grateful to the reviewer for his/her positive evaluation of our work. We are greatly encouraged by the reviewer's comment.

Comment 1

The authors use the expression "carrier consumption" (see for example top of page 4). I find this expression unclear. Can they clarify?

Reply

We thank the reviewer for his/her comment and apologize for our ambiguous expression. Here, we just want to mention that the carrier density decreases as a consequence of stimulated emission. To avoid the ambiguity, we have rephrased this expression to "stimulated carrier recombination".

Comment 2

We can see on fig1b that the laser mode at the A-band edge has the same energy as a D-band mode that crosses the gap. Can this affect the confinement of the laser mode? Can the authors discuss this point?

Reply

We thank the reviewer for his/her important comment. As detailed in our previous paper [R1], for a double-lattice photonic crystal which has reflection symmetry along the u -axis, the band-edge modes can be classified into the following two groups: (1) anti-symmetric modes (A,C), which have electric field vectors that are anti-symmetric about the u -axis, and (2) symmetric modes (B,D), which have electric field vectors that are symmetric about the u -axis. Since mode A and mode D have a different symmetry, they do not interact each other. Therefore, the frequency gap between the two anti-symmetric modes (A and C) determines the confinement of mode A. In the revised manuscript, we have included the above discussion in the explanation of Fig. 1b.

[R1] Inoue, T. et al. General recipe to realize photonic-crystal surface-emitting lasers with 100-W-to-1-kW single-mode operation. Nat. Commun. 13, 3262 (2022).

Comment 3

In figure 1c a width “a” is indicated. What does it correspond to? Is it the same width as “L” in figure 1a?

Reply

We thank the reviewer for his/her comment and apologize for our ambiguous expression. Here, “ a ” denotes the lattice constant of the photonic crystal, and “Larger a ” denotes that the lattice constant gradually increases along the u -axis. To avoid ambiguity, we directly write “Larger lattice constant” in the revised version of Fig. 1c.

Comment 4

The evolution of the pulse width and their repetition frequency as a function of various parameters (injection current,...) is studied theoretically (notably in the supplemental). However, it does not seem obvious to me, from the model, what are the optimum values that can be obtained? What are the physical parameters that limit these values? Can this point be commented on?

Reply

We thank the reviewer for these important comments. As explained in Fig. S3 in Supplementary Information, the pulse width is largely dependent on the parameters of the lattice constant gradation (α_1, β), and the minimum pulse width we obtained in our simulation is 26 ps. We consider that this

value is determined by the following three factors: (1) the differential gain of the active layer $dg(N)/dN$, which determines the speed of the stimulated emission (carrier recombination), (2) the differential refractive index change $dn(N)/dN$ of the active layer, which determines the speed of the band-edge frequency evolution, and (3) the amount of the band-edge frequency change (α_1) inside the device. Although the theoretically achievable minimum pulse width is not clear at this moment, we believe that pulse widths can be further reduced by increasing $dg(N)/dN$ and $dn(N)/dN$ in the active layer. In the revised version of Supplementary Section 3, we have included the above discussion.

The repetition frequency of the proposed device monotonically increases with the injection current as shown in Fig. S2d, owing to the faster carrier accumulation in the active layer after each pulse generation. We can also obtain an arbitrary repetition frequency such as kHz or MHz by using an electrical pulse driver with a current duration of \sim ns, which enables us to extract only the first pulse of the self-pulsation.

Comment 5

The model developed in this article allows to calculate the distribution of charge carriers in the cavity (see for example figure S4 of the supplement). I think it would be interesting to plot this distribution (along u) for the different stages of the pulse propagation (figures 1c-e).

Reply

We thank the reviewer for his/her useful suggestion. According to the reviewer's suggestion, we included the calculated carrier density distribution along the u -axis in the revised version of Fig. 2e, where stimulated-emission-induced carrier-density change is clearly observed.

Comment 6

Apart from simplicity, are there any other advantages to this configuration over traditional Q-switched lasers?

Reply

We thank the reviewer for his/her important comment. As already discussed in the main text with Fig. 4, one advantage of our self-evolving PCSEL, aside from its simplicity, is its large frequency chirping during pulsation, which enables the realization of higher-peak shorter-width optical pulses via pulse compression just by using dispersive optical elements. Moreover, we would like to emphasize that our proposed concept of self-evolving photonic crystals involves not only the above-mentioned practical advantages but also brings new insights into carrier-photon dynamics inside wavelength-scale photonic nanostructures, and it will inspire various areas of fundamental research as well as lead to the discovery of novel photonic devices with new functionalities, as discussed in the

main text.

REVIEWERS' COMMENTS

Reviewer #1 (Remarks to the Author):

The authors made point-to-point responses to all the comments, which are reasonable and clear. I hereby recommend the manuscript be published in its current form.

Reviewer #2 (Remarks to the Author):

My concerns have been perfectly addressed in the revisions and I suggest publication in Nature Com.